# Official International Mahjong: A New Playground for AI Research

**Yunlong Lu [1], Wenxin Li [1,*] and Wenlong Li [2]**

1   School of Computer Science, Peking University, Beijing 100871, China
2   Mahjong International League, 1002 Lausanne, Switzerland
*   Correspondence: lwx@pku.edu.cn; Tel.: +86-10-62753425

**Abstract:** Games have long been benchmarks and testbeds for AI research. In recent years, with the development of new algorithms and the boost in computational power, many popular games played by humans have been solved by AI systems. Mahjong is one of the most popular games played in China and has been spread worldwide, which presents challenges for AI research due to its multi-agent nature, rich hidden information, and complex scoring rules, but it has been somehow overlooked in the community of game AI research. In 2020 and 2022, we held two AI competitions of Official International Mahjong, the standard variant of Mahjong rules, in conjunction with a top-tier AI conference called IJCAI. We are the first to adopt the duplicate format in evaluating Mahjong AI agents to mitigate the high variance in this game. By comparing the algorithms and performance of AI agents in the competitions, we conclude that supervised learning and reinforcement learning are the current state-of-the-art methods in this game and perform much better than heuristic methods based on human knowledge. We also held a human-versus-AI competition and found that the top AI agent still could not beat professional human players. We claim that this game can be a new benchmark for AI research due to its complexity and popularity among people.

**Keywords:** game AI; supervised learning; reinforcement learning; AI competition; Mahjong

## 1. Introduction

Games are born to test and challenge human intelligence and have always been benchmarks and testbeds throughout the history of artificial intelligence. Board games such as checkers [1] and chess [2] were the first to be solved by AI algorithms in the last century. With the boost in computational power and the application of new algorithms, in the last decade, AI systems have achieved superhuman performance in many games that once, only humans were believed to be able to master, from board games such as Go [3–5] and card games such as Texas Hold'em [6–8] to video games such as StarCraft [9], Dota 2 [10], and HoK [11]. These games are all popular among humans, with various annual and seasonal competitions held all over the world. Such popularity has spurred the academic community to put effort into them and develop new algorithms to solve them.

Mahjong is one of the most popular games played in China and has spread worldwide since the early 20th century. Originating from China and with a long history dating back to the 12th century, many regional variants of Mahjong have been developed and are widely played in almost every region of China. It is estimated to have 600 million players all over the world. However, there are so many regional variants in rules that most of the Mahjong communities focus on different regional variants, and there is no competition that covers a majority of players to have enough impact. Because of this, the game has been overlooked by the community of AI research.

Mahjong is a four-player tile-based game with complex scoring rules and rich hidden information, which presents several challenges to AI research. The complexity of the game is much higher than that of Texas Hold'em when measured by the average size of

information sets. Unlike most of the other card games such as poker and bridge, the goal of Mahjong is to form various scoring patterns from the tiles, while other games usually involve a single target such as making a bigger hand or emptying the hand as quickly as possible. Mahjong tiles are equal in their strength, which is different from poker, where cards are ranked from two to Ace and players can have a clear value judgement. The strategy of Mahjong involves choosing between multiple target patterns based on the similarity between the current hand and various patterns, which is hard to estimate but relies more on the aesthetic feeling of the tile combinations. The result of the game is also affected by luck, causing high variance when solved by AI algorithms.

Official International Mahjong is one of the three variants recognized and standardized by the Mahjong International League (MIL) as a set of competition rules called the Mahjong Competition Rules (MCR). This variant consists of 81 different scoring patterns, much larger than other variants, to emphasize the strategic thinking and calculation ability of players. The other two variants are Riichi Competition Rules (RCR) from Japanese Mahjong and Sichuan Bloody Rules (SBR) from Sichuan Mahjong. SBR is dominated by luck instead of strategies and is not suitable for AI research. RCR features a unique set of rules and encourages a more conservative style of play. It has fewer scoring patterns and is easier than MCR, and has been preliminarily solved by the AI system named Suphx built by MSRA [12]. Among those variants, Official International Mahjong is the most suitable variant as a playground for AI algorithms because of the complexity in its scoring rules and the emphasis on strategies instead of luck.

As an effort to promote Official International Mahjong in AI research, we held two Mahjong AI competitions in conjunction with the International Joint Conference on Artificial Intelligence (IJCAI) in 2020 and 2022. Dozens of teams participated in the competitions and used a variety of algorithms to build their agents. These algorithms include heuristic methods based on human knowledge and methods based on deep learning, such as supervised learning and reinforcement learning. The results of the competitions showed that the agents trained by supervised learning and reinforcement learning turned out to perform better than those with heuristic methods. A human-versus-AI competition was also held after the first Mahjong AI competition to challenge professional human players with one of the top AIs. The result showed that the AI agent was still far behind the top human players.

These competitions mark the beginning of Official International Mahjong as a new playground for AI research. While the best AI agents in the competitions still cannot beat the best human players, we believe that AI systems built by new algorithms with more computational power will eventually achieve superhuman performance in this game as long as this game draws enough attention from the community of AI research, which is achieved by more AI competitions. To conclude, our contributions are as follows.

- We are the first to hold AI competitions of Official International Mahjong. We provide the judge program that implements the rules of this Mahjong variant and open source libraries to calculate scoring patterns. We also build match datasets from both humans and top AI agents for further research.
- We innovatively adopt the duplicate format in the evaluation of AI agents to reduce the high variance introduced by the randomness of the game, which is commonly used in top-level human competitions but rarely used in AI competitions.
- We promote Official International Mahjong as a challenge and benchmark for AI research. We summarize the algorithms of AI agents in the two AI competitions and conclude that modern game AI algorithms based on deep learning are the current state-of-the-art methods. We claim that more powerful algorithms are needed to beat professional human players in this game.

The rest of this paper is organized as follows. Section 2 discusses the background of games as benchmarks and testbeds for AI research. An overview of game AI algorithms and a brief introduction to Mahjong are also presented. Section 3 presents a detailed description of the rules of Official International Mahjong. Section 4 introduces the two AI competitions and the human-versus-AI competition we have held of this game, including the platform

and our novel competition format. Section 5 summarizes the algorithms used in the AI agents submitted by the competitors.

## 2. Background

### 2.1. AI and Games

The goal of artificial intelligence is to reach human levels in a variety of tasks. But how can we measure the intelligence level of AI systems against humans? Turing test [13] may be the earliest criterion to test the intelligent behavior of AI systems, but it has been controversial, as more and more AI systems have managed to pass the test by targeted imitation or deception without real intelligence. However, games are created to test and challenge human intelligence, making them excellent benchmarks and testbeds for the evolution of AI. The diversity of games provides a rich context for gradual skill progression to test a wide range of AI systems' abilities. From Deep Blue to AlphaGo, AI systems beating professional players in games with increasing complexity have always been considered major milestones and breakthroughs in the development of AI technologies.

The earliest research on game AI focused on classic board games such as checkers [1], chess [2], and Go [3]. These games are relatively simpler to solve by AI algorithms due to their discrete turn-based mechanics, highly formalized state representation, and fully visible information to all players. However, these games almost exclusively test the strength of different strategies and are not fun enough to be very popular among humans. Nowadays, popular games played by humans are often much more complex, involving randomness and luck to introduce fun, imperfect information between players to introduce deduction and deception, or more complex game elements and multi-player team battles to improve playability. The increasing complexity and diversity of games bring new challenges for AI research, thus promoting the invention of new algorithms to solve them.

In the last decade, AI systems have achieved superhuman performance in many popular games played by humans, thanks to the advances in AI techniques and the boost in computational power. These games include card games such as Texas Hold'em [6–8], which is the most popular card game played in America, and multi-player video games such as StarCraft [9] and Dota 2 [10], which have well-established e-sports events popular all over the world. One important reason these games draw attention to AI research is that these games are famous enough to form communities of players who organize annual or seasonal tournaments both for human players and for AI programs. For example, the World Series of Pokers (WSOP) [14] is a series of poker tournaments held annually since 1970, where over half of the events are variants of Texas Hold'em. Meanwhile, as an effort to develop a system to compare poker agents, the Annual Computer Poker Competition (ACPC) [15] has been held annually since 2006 in conjunction with top-tier AI conferences such as AAAI and IJCAI. ACPC began with only five competitors in the first year but attracted dozens of teams from many countries several years later. Such tournaments provide AI researchers with great platforms to compare and improve their AI agents until more and more powerful agents finally beat professional human players, marking that the game has finally been solved by AI.

### 2.2. Game AI Algorithms

Many AI algorithms have been proposed to build game-playing agents for different games. These algorithms can be divided into three categories [16]. Heuristic methods, also called expert systems, rely heavily on human knowledge, which is the most straightforward way to build a game AI system from scratch. Real-time planning algorithms compute the action to take when each specific game state is actually encountered during gameplay, usually by expanding a search tree from the current state, and are widely used in board games such as chess and Go, where fast responses are not required. Learning algorithms train models in advance to store policies or value functions in the form of model parameters, which are later used in real-time gameplay. Supervised learning and reinforcement learning are the most common learning algorithms in the field of game AI.

There are three basic ways of using explicit human knowledge in heuristic methods. The first way is to incorporate human knowledge in feature extraction. Strategy-related features can be calculated from the raw features of the game state based on the understanding of human players to guide the selection of better actions. The second way is to provide explicit rule-based policies, usually written in an 'if-else' manner. Such policies are formally modeled by decision trees, where each internal nodes are conditions about the game state to choose a child node, and each leaf node is an action label. Some decision trees also have chance nodes, where child nodes are selected based on probabilities. The third way is to create an evaluation function based on human knowledge, to estimate the preference of human players on game states. Such evaluation functions can be further combined with real-time planning algorithms to select the action that potentially leads to better states in the following steps.

In most cases, the state space of a game is so large that it is difficult to have an optimal strategy before the game starts. Real-time planning algorithms can compute the action at each game state in real-time gameplay if the state transition model of the game is known. When an evaluation function of game states is available, the simplest form of planning is to choose an action by comparing the values of the model-predicted next states for each action. Such planning can look much deeper than one step ahead and expand a search tree to evaluate many future states under different sequences of actions, leading to a more far-sighted agent. One of the most famous algorithms is A* [17], which uses an evaluation function to guide the selection of unexplored nodes in the search tree. Minimax [18] is another classical real-time planning algorithm in two-player competitive settings and is widely used in board games, based on the assumption that each player wants to maximize their own payoffs. Monte-Carlo tree search (MCTS) simulates many trajectories starting from the current state and running to a terminal state, and updates the state-action value along the path based on the final score, which is then used to yield better trajectories. This algorithm has achieved huge success in two-player board games and accounts mostly for the progress achieved in computer Go from a weak amateur level in 2005 to a grandmaster level in 2015 [19].

The model trained by learning algorithms is the most fundamental part of building a game AI system and is the focus of game AI research. Supervised learning learns a policy or value model that predicts the action to choose or the estimated value under the current state. It requires lots of labeled data in the form of state–action or state–value pairs, usually collected from human gameplay data or data generated by other game-playing algorithms. Reinforcement learning (RL) models the environment as a Markov Decision Process (MDP) and discovers which actions can lead to higher future rewards by interacting with the environment and trying them out. Traditional RL algorithms deal with single-agent settings and can be divided into value-based methods and policy-based methods, depending on whether they only learn value models or learn policy models as well. PPO is probably the most popular RL algorithm because it empirically has a more stable training process compared to other algorithms, and is widely used in most of the game AI milestones achieved in recent years, such as AlphaStar and OpenAI Five. When combined with proper self-play techniques, single-agent RL algorithms are proved to approach a Nash equilibrium in competitive multi-agent environments [20–22]. In practice, a model pool is usually maintained from which opponents are selected to collect data for training, which has become a general paradigm for applying RL algorithms to multi-agent games.

### 2.3. Mahjong

Mahjong is a game almost as popular in China as Texas Hold'em in America; however, it has not attracted much attention from the community of game AI research. This game and its regional variants are widely played all over the world and estimated to have 600 million players in total, according to an internal report of the Mahjong International League. It is worth noting that Mahjong should not be confused with Mahjong Solitaire [23], which

is a single-player picture-matching game using the same set of tiles. Instead, four-player Mahjong is similar to a Western card game called rummy, using tiles to form patterns instead of cards.

There are many regional variants of Mahjong rules, and here, we present a brief introduction of the basic rules. Mahjong is played with a basic set of 144 tiles with Chinese characters and symbols, though many regional variations may omit some of the tiles. Usually, each player begins with 13 tiles which are not shown to other players. Players draw and discard tiles in turn until they complete a winning hand with a 14th tile. The basic type of winning hand consists of four melds and a pair, while there exist winning hands with several special patterns. Most of the regional variations have some basic rules in common, including the order of play, how a tile is drawn and discarded, how a tile can be robbed from another player, and the basic kinds of melds allowed. Different variations differ mainly on the criteria for legal melds and winning hands, and they can have very different scoring systems and some additional rules. In general, the game of Mahjong requires skill and strategy, as well as a decent amount of luck to win, making it entertaining enough to have many players all over the world.

Among those various regional variants of Mahjong rules, only three of them are recognized and standardized for Mahjong competition by the Mahjong International League, a non-profit organization aiming to promote Mahjong as a mind sport throughout the world. These three variants are recognized mainly because of their popularity and the different emphasis in their strategies. The first of them is Official International Mahjong, founded by the General Administration of Sport in China in July of 1998, known as Mahjong Competition Rules (MCR), which includes a large variety of scoring rules to emphasize strategy and calculation ability, and is widely adopted for Mahjong competition. The second is Riichi Competition Rules (RCR) from Riichi Mahjong, or Japanese Mahjong. This variant features a unique set of rules such as Riichi and the use of dora, encouraging a more conservative style of play. The third is Sichuan Bloody Rules (SBR) from Bloody Mahjong, or Sichuan Mahjong, because this variant is fast-paced and dominated by luck.

From the perspective of game AI, Mahjong presents several challenges for AI algorithms. First, as a four-player game, multi-agent AI algorithms are required to solve it because both competition and cooperation exist between different agents. Second, the private tiles of each agent are not exposed to other agents, and players have to deduce other players' hands by their discarded tiles, making it a game with rich hidden information. The complexity of imperfect-information games can be measured by information sets, defined as the game states that each player cannot distinguish from their own observations [24]. Figure 1 shows the number and average size of information sets of Mahjong. The average size of information sets in Mahjong is much larger than that of Texas Hold'em and another card game popular in China called DouDizhu, making it difficult to solve by CFR-based algorithms, which have achieved success in variants of Texas Hold'em. Last, randomness and luck also contribute to the results of the game, causing high variance and instability in convergence when applying learning algorithms.

Unlike other popular card games such as bridge, Texas Hold'em, and Doudizhu, which also have imperfect information and have been testbeds for AI research, the rules and strategies of Mahjong are unique in the following ways. First, in most card games, each card has a different strength, usually ranking from two to Ace (A). Higher cards can beat lower cards so that each hand can have a clear value judgment in the eyes of players. However, there is no such concept of "strength" of individual tiles in Mahjong, and each Mahjong tile is equal in its role in forming various patterns. The value judgment of each hand is based on the similarity between the current hand and different scoring patterns, which is dominated by the aesthetic feeling of the tile combinations. Second, the strategies of Mahjong involve making the choice between multiple targets of patterns, especially in Official International Mahjong with 81 different scoring patterns, while other card games usually have a single target such as making a bigger hand than others or emptying the hand as soon as possible. These factors make it harder to acquire an optimal strategy in Mahjong,

and even professional players can have quite different value judgments and strategies with the same hand.

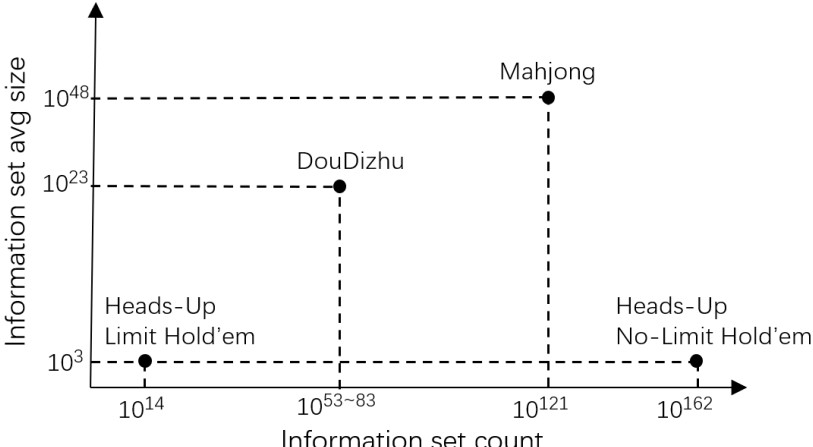

**Figure 1.** The number and average size of information sets of some imperfect-information games.

## 3. Official International Mahjong

As the official variant of Mahjong rules widely used in Mahjong competitions, Official International Mahjong is a better playground for AI research than other Mahjong variants for two reasons. First, unlike most of the regional variants such as SBR where luck dominates the result of the game, Official International Mahjong involves a moderate amount of luck, but mostly strategies and skills. Second, Official International Mahjong has a complex scoring rule with 81 different scoring patterns, many more than in other variants, including RCR. In fact, Official International Mahjong integrates most of the scoring patterns among various regional variants, requiring more sophisticated strategies for human players and indicating higher complexity for AI algorithms. This section introduces the rules of Official International Mahjong in detail.

Official International Mahjong is played with a full set of 144 Mahjong tiles split into 3 categories: suited tiles, honored tiles, and flower tiles, as shown in Table 1. Suited tiles are divided into three suits: Characters, Bamboos, and Dots, each numbered from one to nine. There are 4 identical copies of each tile, totaling 108 suited tiles. Honor tiles are divided into two sets: wind tiles of four directions and dragon tiles of three colors. Like suited tiles, each tile has 4 identical copies, totaling 28 honor tiles. The remaining eight tiles are special flower tiles, each with an artistic rendering of one plant or season, which play a unique role in the mechanics of the game.

When playing the game, four players sit down at their respective positions around the table in the shape of an inverted compass: east is the dealer, the right of the dealer is south, across is west, and the left is north. The position of each player is called their "seat wind". A full match usually consists of four rounds, each representing a "prevailing wind", starting with east. Four games are played each round, with players shifting their seat winds. The seat wind and prevailing wind are important because they affect the scoring of the game, which is further discussed in Appendix A.

When a game starts, all of the tiles are randomly shuffled and stacked as two layers before each player to form a square wall. Tiles are drawn from a specific breaking position of the wall so that the tile wall decreases clockwise, and that position is usually decided by dice rolling. At the start of each game, 13 initial tiles are drawn by each player in a specific order, as shown in Figure 2. Then, players draw and discard one tile in turn in counterclockwise order until some player declares a winning hand, or no one wins before the tile wall runs out, ending in a tie.

This regular order of play can be interrupted in four cases. In the first case, whenever a player draws a flower tile, it should be exposed and put aside. The player has to draw

the last tile of the wall as a replacement so that they still have 14 tiles before they discard any tile. This also happens when dealing the initial 13 tiles, and can happen successively in a player's turn if they draw another flower tile as the replacement.

**Table 1.** A full set of 144 Mahjong tiles consists of 108 suited tiles, 28 honor tiles, and 8 flower tiles.

| Suited Tiles | | Numbers | | | | | | | | |
|---|---|---|---|---|---|---|---|---|---|---|
| | | One | Two | Three | Four | Five | Six | Seven | Eight | Nine |
| Suits | Characters | 一萬 | 二萬 | 三萬 | 四萬 | 伍萬 | 六萬 | 七萬 | 八萬 | 九萬 |
| | Bamboos | | | | | | | | | |
| | Dots | | | | | | | | | |

| Honor tiles | Winds | | | | Dragons | | | | | |
|---|---|---|---|---|---|---|---|---|---|---|
| | East | South | West | North | Red | Green | White | | | |
| | 東 | 南 | 西 | 北 | 中 | 發 | | | | |

| Flower tiles | Plants | | | | Seasons | | | | | |
|---|---|---|---|---|---|---|---|---|---|---|
| | Plum blossom | Orchid | Daisy | Bamboo | Spring | Summer | Fall | Winter | | |
| | 梅 1 | 蘭 2 | 菊 3 | 竹 4 | 春 1 | 夏 2 | 秋 3 | 冬 4 | | |

In the second case, when a player discards a tile, any other player may steal the tile to complete a meld. Melds are groups of tiles within players' hands, which are essential components to form a winning hand. There are three kinds of melds: pungs, kongs, and chows. A pung is three identical tiles, either suited tiles or honor tiles. A kong is four identical tiles, but only counts as a set of three tiles. A chow is three suited tiles of the same suit in a consecutive numerical sequence. When a meld is formed by stealing another player's discard, it is put aside and exposed to all players; otherwise, it remains concealed. A player can only steal a tile from the player left to them to form a chow, but from any player to form a pung or a kong. If multiple players try to steal the same discard, the priority of pung and kong is higher than that of chow. After the player steals the tile to complete a pung or a chow, they become the next player to discard a tile because there are currently 14 tiles in their hand. However, since a kong only counts as three tiles, the player who steals a tile to complete a kong has to draw another tile from the end of the wall before they discard their tile.

In the third case, a player can also declare a kong upon drawing a tile. There are two different ways to complete such kongs. In the first case, the player can declare a concealed kong if they hold four identical tiles in their hand, which do not necessarily include the tile they just drew. These four tiles are put aside but remain concealed from other players. In the second case, if the player has an exposed pung and holds the fourth tile, they can promote the pung to a kong by adding the fourth tile. In both cases, the player has to draw another tile from the end of the wall before they discard their tile.

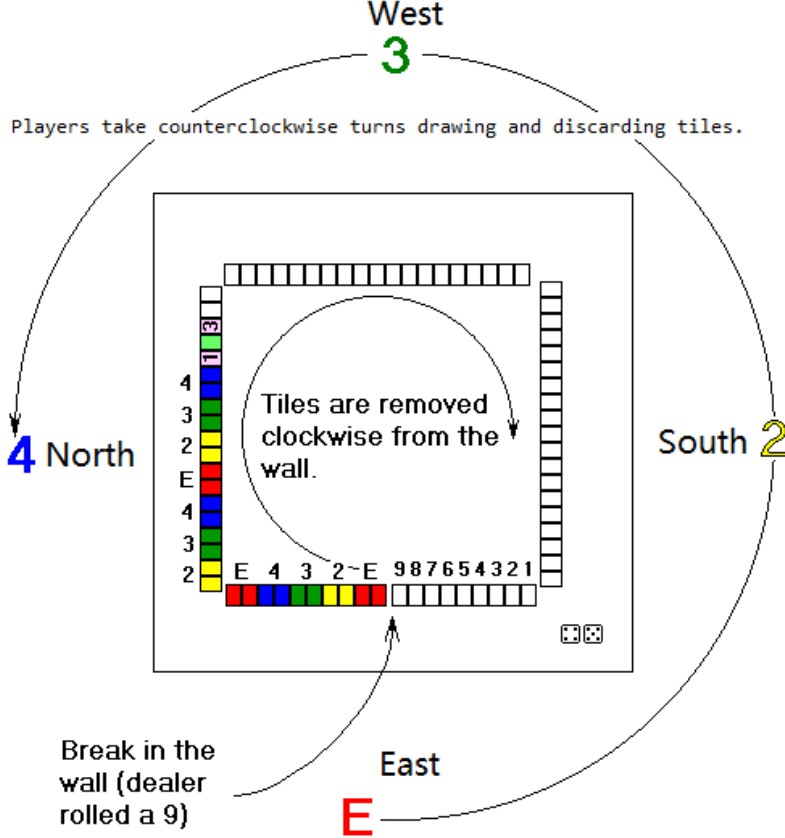

**Figure 2.** The seat positions of players and the initial configuration of the two-layer tile wall. The tile wall decreases clockwise from a breaking point, and the colors indicate the order of dealing the initial 13 tiles of each player. Then, players draw and discard tiles in counterclockwise order. Image modified from [25].

In the fourth case, a player can declare a winning hand and end the game. All their tiles are then exposed to others for a validity check. A winning hand consists of 14 tiles. Since players always have 13 tiles in their hand during play (flower tiles are not counted), they must win by either drawing a tile or seizing another player's discarded tile, making a 14-tile winning hand. There is a special case when another player adds a fourth tile to promote a pung to a kong; a player can rob it if they can make a winning hand with it.

A winning hand has to satisfy specific patterns. The basic form of a winning hand consists of four melds, either exposed or concealed, and a pair of identical tiles called the eye. The hand is then compared to a variety of scoring patterns to count a so-called "*fan*" value. There are 81 different scoring patterns in total, and the detailed list can be found in Appendix A. Each matched pattern is worth some *fan*s, and they are summed up as the *fan* value of this hand. A player can only win when their hand is worth no less than eight *fan*s. There are also some special forms of winning hands, which match scoring patterns without four melds and an eye. All forms of winning patterns are listed in Table 2. The final scores of each player depend on the winner's *fan* value and the provider of the 14th winning tile. Specifically, if the winner makes a winning hand of $x$ *fan*s by drawing a tile themselves, they receive $8 + x$ points from the other three players. Instead, if the 14th winning tile comes from another player, either discarded or added to the promoted pung, the winning player receives $8 + x$ points from the provider of this tile, and only 8 points from the other two players.

**Table 2.** Five forms of winning patterns in Official International Mahjong. A hand of basic winning pattern has to meet eight *fan*s to win, while other patterns are worth more than eight *fan*s themselves.

| Winning Patterns | Explanation with an Example |
| --- | --- |
| Basic pattern | Four melds and a pair, with *fan* value no less than 8. |
| Thirteen orphans | 1 and 9 of each suit, one of each wind, one of each dragon, and one duplicate tile of any. |
| Seven pairs | A hand with seven pairs. |
| Honors and knitted tiles | A hand of 14 tiles from these 16 tiles: number 1, 4, 7 of one suit; number 2, 5, 8 of second suit; number 3, 6, 9 of third suit; and all honor tiles. |
| Knitted Straight | Number 1, 4, 7 of one suit, number 2, 5, 8 of second suit, number 3, 6, 9 of third suit, plus a meld and a pair. |

Since a full match of Mahjong consists of four rounds with different prevailing winds and each round is composed of four games with different seating positions, the final ranking of players depends on the total scores of all sixteen games. Although the scoring of each game is zero-sum, where players are strictly competitive with each other, it is common to cooperate with some players based on the cumulative scores to secure one's final ranking and adopt some different strategies, especially in the last few games of a match.

## 4. AI Competitions

In an effort to unlock the potential of modern game AI algorithms in Official International Mahjong and promote it as a playground for AI research, we held two AI competitions in 2020 and 2022 in conjunction with the International Joint Conference on Artificial Intelligence (IJCAI). Dozens of teams from universities and companies participated in the competitions, and they contributed their wisdom and submitted their AI programs to play Mahjong with a variety of algorithms. We adopted the duplicate format in these competitions under the guidance of the Mahjong International League, which greatly reduces the variance in the evaluation of AI agents. We also organized a human-versus-AI competition in the beginning of 2021, where two professional human players were invited to challenge the top AIs from the AI competition in 2020. This section presents an introduction of these competitions.

### 4.1. Botzone

Both Mahjong AI competitions were held based on Botzone [26], an online multi-agent competitive platform built by our lab. The platform is designed to provide a universal interface to evaluate AI agents in different programming languages in a variety of games, including traditional board games such as Gomoku, Ataxx, chess, and Go, card games such as DouDizhu and Mahjong, and modified versions of Atari games such as Pac-Man and Tank. Users can upload their programs to the platform, called bots, which can play against any other bots of the same game as long as they follow the input–output protocol specified by the platform. The platform also supports games between human players and bots, or only human players.

Apart from creating isolated games to evaluate the performance of agents, Botzone has an Elo ranking system that constantly runs games between randomly selected agents and

calculates the real-time Elo scores of each bot, maintaining a ladder list of each game. Additionally, Botzone also supports creating groups to hold individual tournaments, with multiple rounds of games played between a fixed set of agents under predefined formats. This tournament system to evaluate AI agents makes Botzone a strong platform to support both AI research and education. Over the years, dozens of courses have used this platform to hold competitions in their course project to write AI agents for various games. The platform has so far accumulated more than 160 thousand AI agents in over 40 games, with more than 50 million games played.

*4.2. Competition Format*

As a competition to evaluate the performance of multiple agents, it is important that the final ranking can reveal the actual strength of different agents. However, Mahjong is a four-player imperfect-information game with a high degree of randomness in dealing tiles, making it hard to design the competition format with an accurate ranking. In both Mahjong AI competitions, we applied a combination of Swiss rounds and the duplicate format as the competition format, which reduces the variance in individual games as much as possible. The illustration of the format is shown in Figure 3.

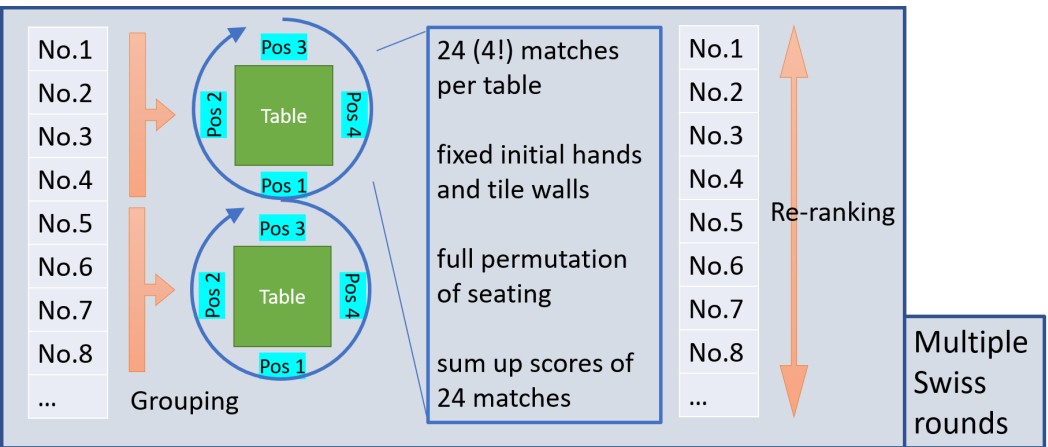

**Figure 3.** The competition format of Mahjong AI competitions, a combination of a Swiss-round system and duplicate format.

Duplicate format is a popular technique to mitigate the factor of luck and reduce variance in games. It was initially applied in the game of bridge and was later used heavily in poker competitions, including ACPC. The basic idea is to play multiple games with players taking different seats while keeping the same outcomes of all random events in each game, including the cards or tiles dealt throughout the game. In two-player games such as Heads-Up No-Limit Hold'em (HUNL), each game is played twice with the same set of cards for each seat, but with different player seating. This can significantly mitigate the factor of luck because even if a player has a good hand and wins the game, their opponent will eventually have the same hand in another game. However, Mahjong is a four-player game where not only the tiles dealt to each player contribute to the result of the game, but the order of players in drawing and discarding the tile also matters. When applying duplicate format to Mahjong, the public tile wall has to be split into four parts for each seat, and each player draws tiles only from their own part, including their initial hands. The same tile wall is used for a total of 24 games, where the players change their seats in all possible permutations. The scores of these games are summed up as the score of one "duplicate game". In this way, not only is the factor of luck from the tile wall mitigated, but the influence of each player to other players such as stealing tiles can also be reduced. It is worth noting that though Suphx built by MSRA is reported to beat top human players in Riichi Mahjong, no duplicate format was used in the evaluation, and their results may suffer from high variance.

As Mahjong is a four-player game, competition formats such as round robin cannot be applied because there are too many possible combinations of players. Instead, the competitions use Swiss round as the high-level format, with each round pitting the closest-ranked players against each other. The initial ranking of players is randomly decided before the first round. In each round, four players are grouped to play 24 games of duplicate format. The raw scores of each game are summed up to decide the rankings of each player in this "duplicate game" to assign ranking points of 4, 3, 2, or 1 to each player. The players are then re-ranked by their accumulated ranking points. The typical Swiss-round format has a relatively small number of rounds, usually the logarithm of the number of players. However, considering the randomness and imperfect information in Mahjong, it takes hundreds of Swiss rounds to achieve a stable ranking of players, especially when they are close in level of play.

Additionally, to further mitigate the factor of luck and reduce the variance in each game, the flower tiles in Mahjong, i.e., plant tiles and season tiles, are excluded in both Mahjong AI competitions and the human-versus-AI competition, because flower tiles only affect the final scores of each player after someone makes a winning hand, and are not involved in the strategy of either discarding or melding. The design of flower tiles makes it a random perturbation term in the accurate evaluation of agents, though it can bring fun to human players as a factor of luck. In conclusion, the competition format of both Mahjong AI competitions combined Swiss rounds, duplicate format, and the exclusion of flower tiles to make the ranking as accurate as possible.

*4.3. Mahjong AI Competition*

We held two Mahjong AI competitions in 2020 and 2022. Both of them were divided into 3 rounds: the qualification round to determine the top 16, the elimination round to determine the top 4, and the final round to determine the champion. The participants need to upload their bots before the qualification round and the elimination round, while the final round uses the same bots as the elimination round.

The first Mahjong AI competition [27] was held in conjunction with IJCAI 2020, which was postponed in the second year as a virtual conference due to the COVID-19 pandemic. Thirty-seven teams submitted their agents in the qualification round on 31 November, where three tournaments were held in a week, each running six Swiss rounds and each round comprising four "duplicate games" of twenty-four matches. The results of the three tournaments were weighted proportionally to calculate the ranking of the qualification round. The top 16 teams joined the elimination round on 1 January 2021, which comprised a single tournament of 96 Swiss rounds, each comrising $4 \times 24$ games in duplicate format. The final round was held later, with 128 Swiss rounds, to provide a more accurate ranking of the top four agents. We organized a symposium during the IJCAI 2020 session, and the top 16 teams all made oral presentations to discuss their methods. Some of the teams also submitted papers to introduce their algorithms in detail.

The second Mahjong AI competition [28] had a similar schedule to the first one, and was held as a competition session at IJCAI 2022. The qualification round was on 22 May, the elimination round was on 3 July, and the final round was on 4 July. The competition format was slightly different from the competition in the first year. First, only one tournament was held in the qualification round instead of three. Second, each Swiss round consisted of only 1 duplicate game with 24 games of the same till wall, but more Swiss rounds were used to ensure that each agent still played hundreds of duplicate games or thousands of individual games in total. Most of the top 16 teams gave their presentations to introduce their algorithms in the symposium held on 28 July during the session at IJCAI 2022. The schedules of both competitions are summarized in Table 3, and the list of winning teams can be found on the homepages of the competitions.

**Table 3.** Summary of the schedule of the two Mahjong AI competitions.

| Year | Qualification Round | | | Elimination Round | | | Final Round | | | Dataset |
|---|---|---|---|---|---|---|---|---|---|---|
| | Time | Teams | Rounds | Time | Teams | Rounds | Time | Teams | Rounds | |
| 2020–2021 | 31 November | 37 | $3 \times 6 \times 4$ | 1 January | 16 | $96 \times 4$ | 6 January | 4 | $128 \times 4$ | Human data |
| 2022 | 22 May | 25 | 128 | 3 July | 16 | 128 | 4 July | 4 | 512 | AI data |

Since Official International Mahjong has a complex scoring system involving 81 scoring patterns and some principles regarding whether each pattern is counted in various cases, it is very difficult and error-prone for the competitors to write code to calculate scores when playing the game. We provided an open source library [29] to calculate the *fan* value of each hand, with two versions in Python and C++. Additionally, we built two Mahjong datasets for the competitors to use in these competitions. Specifically, the dataset provided in the first competition consists of about half a million matches of human players from an online Mahjong game platform, which belongs to another variant of Mahjong with different scoring rules, but shares the same basic rules and some of the most common scoring patterns. The dataset provided in the second competition consists of over 98 thousand self-play matches generated on Botzone by the top agents of the first competition, which has much higher quality than the previous dataset in the strength of strategies. Many teams used the provided dataset for algorithms based on machine learning, which is further discussed in Section 5.

### 4.4. Human-versus-AI Competition

The human-versus-AI competition was held on 30 January 2021, shortly after the final round of the first Mahjong AI competition. We invited two professional human players to the competition to compete against AI programs. The human players were Wenlong Li, the winner of the Japan MCR Championship (2016) who is ranked in the top three of the MIL master points system (2022), and Zhangfei Zhang, the winner of the China MCR Championship (2010). The AI program was called Kima, which was the agent ranked third in the first Mahjong AI competition.

We also adopted the duplicate format in this competition, where multiple games are played with the same tile wall but different player seatings. However, the application of duplicate format to competitions with human players is different from that with AI players, because a human player can memorize the tile wall if the same tile wall is used for multiple games. Instead, the competitors sat in the same position at different tables and were faced with the same tile wall, while the other three players on each table were another AI program as a fixed baseline player, which was chosen as the agent ranked fourth place. A total of 16 games were played in the competition, with a full round of prevalent and seat winds of the competitors. The scores of each player were summed up to decide the final ranking. This competition format can mitigate the factor of luck introduced by both the tile wall and the opponents, since the tile wall and the opponents of each player are all the same.

The competition results are shown in Table 4. Due to the use of the duplicate format to control variance, three competitors achieved the same score in about half of the games, but the two human players tended to perform better than the AI programs in the remaining games, resulting in a huge difference in the total scores between the human players and the AI agents. In an interview after the competition, Li claimed that the AI agent was still not flexible enough to handle some cases. It can be concluded that the AI ranked third place in the first Mahjong AI competition is still a far cry from professional human players, and Official International Mahjong can be a good benchmark and testbed for AI research, calling for more powerful algorithms.

**Table 4.** The scores of each game in the human-versus-AI competition.

| Player | East Wind | | | South Wind | | | | | West Wind | | | | North Wind | | | Total |
|---|---|---|---|---|---|---|---|---|---|---|---|---|---|---|---|---|
| Li | −22 | 32 | 51 | −19 | 54 | 99 | −8 | −17 | 34 | −8 | 34 | −8 | 34 | −27 | −8 | −16 | 205 |
| Zhang | −22 | 60 | 54 | −19 | 54 | −8 | −8 | −30 | 37 | 33 | 66 | −8 | 35 | −16 | −8 | −16 | 204 |
| Kima (AI) | −22 | 60 | −8 | −19 | 54 | −8 | −8 | −28 | −8 | −8 | 66 | −8 | 35 | −8 | −8 | −16 | 66 |

## 5. AI Algorithms

This section summarizes the algorithms used in the AI agents in the two Mahjong AI competitions based on the presentations given by the competitors. Though the details of the algorithms vary, they can be divided into three categories: heuristic methods, supervised learning, and reinforcement learning. An overview of the methods of the top 16 agents in both competitions is illustrated in Figure 4.

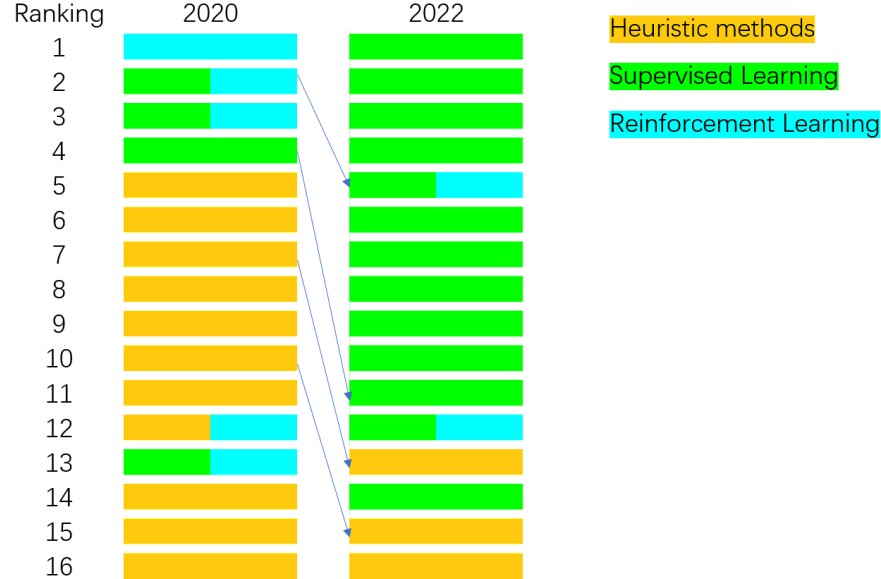

**Figure 4.** The category of algorithms used in the top sixteen agents in the two Mahjong AI competitions. The arrows indicate that the agents were uploaded by the same team who participated in both competitions.

### 5.1. Heuristic Methods

Most of the teams in the first competition built their AI agents with heuristic methods, which rely heavily on human knowledge. For a human player, the goal of Mahjong is to make a winning hand as soon as possible by choosing the right tile to discard and stealing other players' discarded tiles to make melds when necessary. The minimal number of tiles to replace from the current hand to a winning pattern is defined as the "shanten" value, which is an important concept in the strategies of humans to measure their distance from winning. Some heuristic methods simply calculate the current shanten value by expanding a search tree, trying all possible tiles it can replace until it forms a winning pattern. Upon finding the shortest path to a winning pattern, the agent just keeps the useful tiles and discards a tile from those to be replaced until the shanten value drops to zero, waiting for the 14th tile to win. However, such shanten values only reflect the distance between the current hand and a winning pattern, but the *fan* value of the winning pattern may not be enough to win; that value must be no less than eight. Since the *fan* value depends on a wide variety of scoring patterns that not only describe the tiles in the hand, but also the exposed or concealed state of each meld and some properties of the 14th

tile, including these states in the search tree would make it too large to be traversed in limited time.

Instead, human players not only try to reduce the shanten value but also think about the scoring patterns they want to make in advance. Since there are five different winning patterns in Official International Mahjong, and the strategies differ when making different patterns, some heuristic methods try to decide on a pattern to make based on the current hand and move to the pattern according to detailed strategies. These methods usually choose the pattern according to predefined rules based on the experience of human players. For example, human players tend to make the "Seven Pairs" pattern if there are five pairs in the current hand, and make the "Thirteen Orphans" pattern if there are no less than 11 terminal tiles and honor tiles. Some AI agents even consider dozens of scoring patterns as targets, choosing the target based on predefined conditions, and use different strategies for each pattern, making a complex behavior tree.

After deciding which winning pattern or scoring pattern to make, more heuristics can be applied based on the targeted pattern. Some of the AI agents calculate the winning probability by the probability of occurrence of each tile based on visible information. The action is chosen based on a combination of reducing shanten values and maximizing the winning probability. All of these rely heavily on human expertise. Learning-based models can also be embedded in the behavior tree as the detailed strategies for some patterns. For example, the AI agent ranked 12th in the first competition integrates a model learned by reinforcement learning into a behavior tree as the detailed strategy of the basic winning pattern, while the other patterns use heuristic strategies.

In conclusion, heuristic methods rely heavily on human experience and simulate the way human players play the game. Search-based algorithms are used to calculate the shanten value or approximate the winning probability to guide the choice of actions. Most of them also use predefined rules to decide the target pattern to make, forming a behavior tree with more detailed strategies embedded to handle each targeted pattern. The performance of these agents is mainly restricted by the strength of human heuristics, and the behavior trees designed by better human players tend to be more complex and have better performance.

*5.2. Supervised Learning*

Supervised learning was also widely used, especially in the second Mahjong AI competition, probably because of the availability of high-quality datasets. The basic idea of supervised learning is to train a policy model to predict the action to choose under the current observation from the match dataset in advance, and directly use the model to compute an action during the gameplay. Since the policy model is learned directly to clone the behavior in the match dataset, the performance of the model depends on the quality of the dataset. Despite the low quality of the dataset provided in the first competition, which consists of matches of human players of different levels in another Mahjong variant, agents trained by supervised learning still outperform those using heuristic methods, indicating the huge potential of deep learning algorithms on this game. More participants turned to supervised learning in the second competition, making up most of the top 16 agents.

The supervised models of different teams differ mainly in three components: the design of the features, the design of the network model, and the data preprocessing scheme. Most teams designed the features as all visible information from the current player, including the current hand, the exposed melds and their types for each player, the discarded tiles of each player, the prevalent wind and the seat wind, and the number of unseen tiles. Some teams missed some of the features or caused information loss by combining some of the features, such as ignoring the order of the discarded tiles. Meanwhile, these features were encoded differently as tensors by different teams. The most basic method is to treat all of these features as vectors and stack fully connected layers as the network model. While such encoding can retain all useful information, it overlooks the spatial relationship between the Mahjong tiles, since most of the scoring patterns involve special patterns of

tiles such as pungs and shifted chows. Suphx [12], an AI system built by MSRA to play Riichi Mahjong, encodes the hand as 4 channels of images of $34 \times 1$, and uses 1-dimensional convolutional neural networks (CNN) to extract high-level features from the multi-channel images. Inspired by and improved from the design of Suphx, many teams in the competition encoded the tiles as 4 channels of images of $4 \times 9$ to better capture the spatial relationship between suited tiles of the same number. The encoding of both schemes is illustrated in Figure 5.

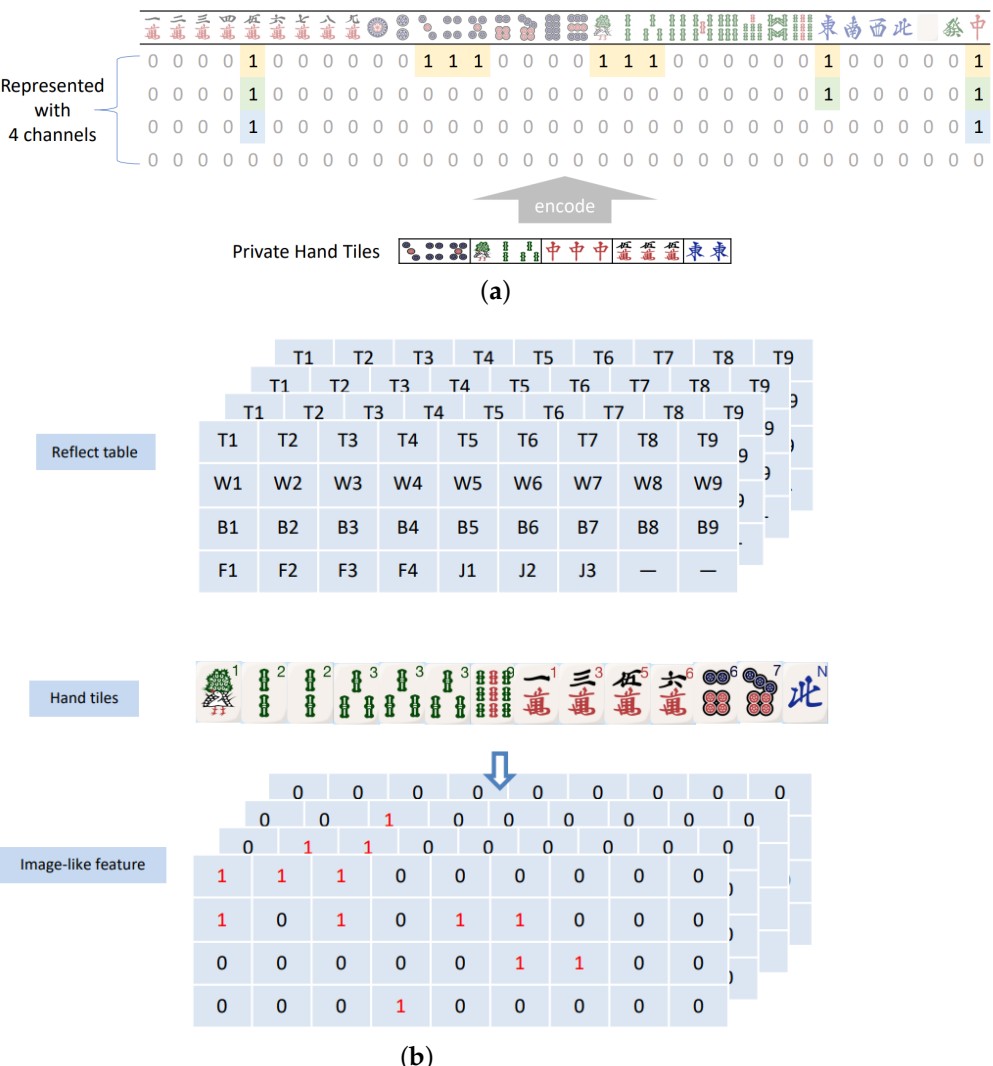

**Figure 5.** The encoding of features in Suphx and some AI agents submitted by the competitors. (**a**) Tiles encoded as 4 channels of images of $34 \times 1$. Image from [12]. (**b**) Tiles encoded as 4 channels of images of $4 \times 9$. Image from [30].

While the designs of features differ in most of the AI agents using supervised learning, the action space of different agents is quite similar, including winning, discarding, and different types of melding. Most of the teams used variants of CNN as the network model to predict the action, a method that is widely used in the field of computer vision to extract features from images due to its strong capability to capture spatial patterns. Specifically, the agents in the first Mahjong AI competition mainly use Resnet [31] as the CNN model, while other variants of CNN such as ResNeXt [32] can also be seen in the agents submitted in the second competition. These variants of CNN adopt deeper layers by adding shortcut connections, or use kernels with more complex structures to increase the ability to extract high-level features.

Before training the network, the given dataset of matches is first preprocessed into millions of state–action pairs. The schemes of data preprocessing of these teams are also different from each other. Some teams simply took all of the actions made by each player and the corresponding observation of the player as the states in these matches. A few teams only used the state–action pair of the winning player of each match, based on the assumption that the winner's strategy was better than that of the other players, especially for the dataset of human matches provided in the first competition. Some teams also applied data augmentation, based on the fact that some tiles are symmetric in their roles to form scoring patterns. For example, three types of suits can be exchanged arbitrarily, and the suited tiles of numbers one to nine can also be reversed as nine to one without breaking most of the scoring patterns, except for some rare patterns such as "All Green" and "Reversible Tiles". Different wind tiles and dragon tiles can also be exchanged as long as the prevailing wind and the seat wind change correspondingly. This augmentation can significantly increase the amount of data and improve the generalization ability of the network model.

*5.3. Reinforcement Learning*

The top three teams in the first Mahjong AI competition all used reinforcement learning (RL) to train their agents. Different from supervised learning which requires a large amount of game data to clone the behavior of the other players, RL deals with control problems and learns how to choose the right action to lead to higher future payoffs by interacting with other players and trying the actions out. Traditional reinforcement learning models the environment as a Markov Decision Process (MDP) and can only deal with single-agent problems. When combined with proper self-play schemes, it has been proved [22,33] that single-agent RL algorithms can approach a Nash equilibrium in competitive multi-agent environments, and in recent years, such algorithms have achieved good performance in various multi-agent games such as StarCraft [9], Dota 2 [10], and Texas Hold'em [34].

All of the top three teams used a similar paradigm of reinforcement learning, combining the algorithm of Proximal Policy Optimization (PPO), a distributed actor–critic training framework, and the self-play of the latest model to collect training data. PPO [35] is a policy-based RL algorithm which is more stable than other algorithms, such as REINFORCE [36] and A3C [37], because it clips the ratio of action probabilities between new models and old models. The network model is a joint policy–value network consisting of a CNN-based backbone to extract features, a policy branch to predict the actions, and a value branch to provide the estimated expected payoff under the current observation. The distributed training framework consists of a learner, a replay buffer, and many actors distributed across multiple machines, as shown in Figure 6. The actors use the latest model to generate self-play matches and store those data in the replay buffer. The learner samples data from the replay buffer to update the current model and broadcast the latest parameters to each actor. This distributed training framework can be easily scaled to an arbitrary amount of computational resources. Since reinforcement learning requires a large amount of data generated by self-play matches, especially in games with high variance such as Mahjong, which involves randomness in dealing tiles, the quality of models depends mainly on the computational resources used. In fact, the top three teams in the first competition were all from the industrial community, and hundreds of CPU cores were used in their training, in spite of the fact that they also adopted various tricks to stabilize the training.

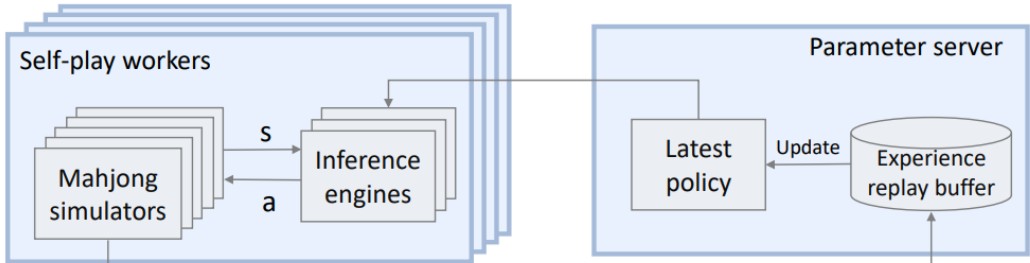

**Figure 6.** The distributed actor–critic training framework used by the competitors.

Though these teams shared a similar training paradigm, there were also some differences in their RL implementations. Some teams used supervised learning to train an initial model from the given datasets or matches generated by heuristic methods in advance, and ran reinforcement learning based on the initial model to speed up the training in the early phase. The agent ranked first in the first competition further adopted the technique of reward shaping and perfect information distillation. Specifically, they did not directly use the final score of the game as the reward, as other teams did, but clipped it to a smaller range to mitigate the randomness and high variance and stabilize the training. Perfect information distillation [38] is another technique to deal with games with imperfect information. While the policy network only takes the observation of the current player as the input, the value network also takes invisible information such as other players' hands and the remaining tile walls as the input to better predict the value of the current game state.

## 6. Conclusions

Throughout the history of AI research, games have always been benchmarks and testbeds because they provide a rich context to test a wide range of abilities of AI systems. The most popular games played by humans have undoubtedly promoted the development of AI algorithms, in that AI systems achieving superhuman performance in these games have always been considered milestones of AI technologies. Mahjong, one of the most popular games played in China, presents challenges to AI research due to its complex scoring rules and rich hidden information, but it has been overlooked by the community of AI research because there are too many regional variants to form a joint community of players. This paper provides a detailed description of the rules of Official International Mahjong, the variant used in most official competitions due to its complex scoring rules and the emphasis on strategies. By holding two AI competitions under our novel competition format, which combines Swiss rounds and the duplicate format to reduce variance, and comparing the methods of the top teams, we promote this game as a new benchmark for AI research and summarizing the state-of-the-art algorithms on this game. We conclude that supervised learning and reinforcement learning perform better than heuristic methods based on human knowledge. We also held a human-versus-AI competition, which showed that the top AI agent still cannot beat professional human players. We believe that this game can be a new playground for game AI research and promote the development of multi-agent AI algorithms in the setting of imperfect information. As Mahjong is a popular game among humans, the strategies of AI agents in playing the game can also bring inspiration to human players, because even professional players can have quite different strategies in some situations.

**Author Contributions:** Resources, W.L. (Wenlong Li); Writing—original draft preparation, Y.L.; Writing—review and editing, W.L. (Wenxin Li). All authors have read and agreed to the published version of the manuscript.

**Funding:** This research was funded by Science and Technology Innovation 2030—"The New 34 Generation of Artificial Intelligence" Major Project No.2018AAA0100901, China, and by Project No.2020BD003 supported by PKU-Baidu Fund.

**Institutional Review Board Statement:** Not applicable.

**Informed Consent Statement:** Not applicable.

**Data Availability Statement:** The data presented in this study are available on request from the corresponding author. The data are not publicly available due to privacy restrictions.

**Conflicts of Interest:** The authors declare no conflict of interest. The funders had no role in the design of the study; in the collection, analyses, or interpretation of data; in the writing of the manuscript; or in the decision to publish the results.

## Appendix A. Scoring system in Official International Mahjong

This section describes the scoring system in Official International Mahjong in detail, including the scoring principles and the full list of scoring patterns.

### Appendix A.1. Scoring Principles

There are 81 different scoring patterns in Official International Mahjong, many more than those in other variants. When a player declares a winning hand, it is compared to all of these patterns, and several patterns may match or even occur more than once. Most of the patterns specify part or even all of the tiles in the hand, but there are also some patterns describing how the winning hand is completed, i.e., the property of the winning tile. Each scoring pattern is worth a *fan* value, and the player only wins when their hand is worth more than eight *fan*s. When counting the *fan* value of a winning hand, the basic rule is to sum up the *fan* values of all matched patterns. However, in many cases, the matched patterns may overlap in their descriptions of the hand, and some patterns are not counted in. The following principles are followed when choosing the right patterns to sum up.

- The Non-Repeat Principle: When a scoring pattern is inevitably implied or included by another pattern, the pattern with a lower *fan* value is not counted.
- The Non-Separation Principle: After combining some melds to match a scoring pattern, these melds cannot be separated and rearranged to match another pattern.
- The Non-Identical Principle: Once a meld has been used to match a scoring pattern, the player is not allowed to use the same meld together with other melds to match the same pattern.
- The High-versus-Low Principle: When there are multiple ways to break the hand to match different sets of scoring patterns, the way with the highest total *fan* value is chosen.
- The Account-Once Principle: When a player has combined some melds to match a scoring pattern, they can only combine any remaining melds once with a meld that has already been used to match other patterns.

### Appendix A.2. Scoring Patterns

All of the scoring patterns are listed below, grouped by their corresponding *fan* value. A pattern with higher *fan* value tends to be more difficult to obtain.

**Patterns worth 88 *fan*s**

**(1) Big Four Winds:**  A hand with pungs or kongs of all four winds.
Implied patterns are not counted: "Big Three Winds", "All Pungs", "Seat Wind", "Prevalent Wind", "Pungs of Terminals or Honors".
Example: 

**(2) Big Three Dragons:**  A hand with pungs or kongs of all three dragons.
Implied patterns are not counted: "Dragon Pung", "Two Dragon Pungs".
Example: 

**(3) All Green:** A hand consisting of only green tiles, i.e., 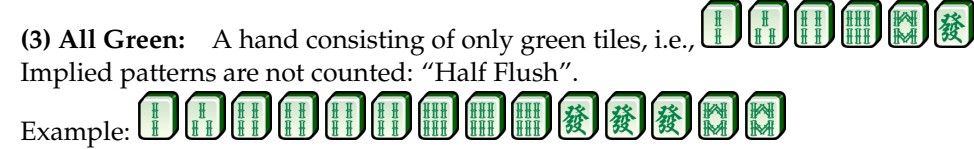. Implied patterns are not counted: "Half Flush".

Example: 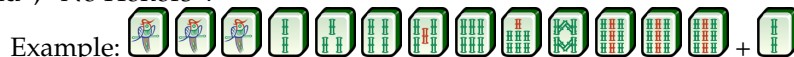

**(4) Nine Gates:** First collect 13 number tiles 1112345678999 of one suit with no exposed melds. The 14th tile can be any tile of the same suit.

Implied patterns are not counted: "Full Flush", "Fully Concealed Hand", "Concealed Hand", "No Honors".

Example: 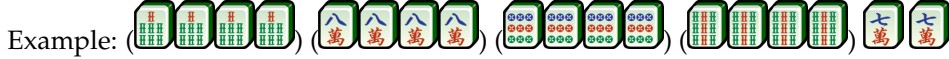

**(5) Four Kongs:** A hand with four kongs, regardless of whether they are exposed or concealed.

Implied patterns are not counted: "Melded Kong", "Concealed Kong", "Two Melded Kongs", "Two Concealed Kongs", "Three Kongs", "All Pungs", and "Single Wait".

Example: (🀫🀫🀫🀫) (🀎🀎🀎🀎) (🀡🀡🀡🀡) (🀘🀘🀘🀘) 🀍🀍

**(6) Seven Shifted Pairs:** A hand of seven pairs in the same suit with consecutive numbers.

Implied patterns are not counted: "Seven Pairs", "Full Flush", "Fully Concealed Hand", "Single Wait".

Example: 🀙🀙🀚🀚🀛🀛🀜🀜🀝🀝🀞🀞🀟🀟

**(7) Thirteen Orphans:** One of each terminal tile (number one and nine of each suit), one of each honor tile, plus one duplicate tile of any of those thirteen tiles.

Implied patterns are not counted: "All Types", "Fully Concealed Hand", "Concealed Hand", "Single Wait".

Example: 🀐🀘🀇🀏🀙🀡🀀🀁🀂🀃🀄🀅🀆🀆

**Patterns worth 64 *fan*s**

**(8) All Terminals:** A hand consisting of only terminal tiles, i.e., number 1 or 9 of each suit.

Implied patterns are not counted: "No Honours", "Pung of Terminals or Honors", "Mixed Double Pung", "All Pungs".

Example: 🀇🀇🀇🀙🀙🀙🀏🀏🀏🀐🀐🀐🀡🀡

**(9) Little Four Winds:** A hand with melds of three winds and a pair of the other wind.

Implied patterns are not counted: "Big Three Winds", "Pung of Terminals or Honors".

Example: 🀀🀀🀀🀁🀁🀁🀂🀂🀂🀃🀃🀐🀐🀐

**(10) Little Three Dragons:** A hand with melds of two dragons and a pair of the other dragon.

Implied patterns are not counted: "Two Dragon Pungs" and "Dragon Pung".

Example: 🀄🀄🀄🀅🀅🀅🀆🀆🀡🀡🀡🀐🀐

**(11) All Honors:** A hand consisting of only honor tiles.

Implied patterns are not counted: "All Pungs", "Pung of Terminals or Honors".

Example: 🀅🀅🀅🀄🀄🀄🀂🀂🀂🀃🀃🀃🀀🀀

**(12) Four Concealed Pungs:** A hand with four concealed pungs or kongs.

Implied patterns are not counted: "All Pungs", "Three Concealed Pungs", "Two Concealed Pungs", "Fully Concealed Hand", "Concealed Hand".

**(13) Pure Terminal Chows:** A hand with two chows of number 1, 2, 3, two chows of number 7, 8, 9, and a pair of number 5, all in the same suit, i.e., 12312378978955 in one suit. It is not treated as seven pairs, even if all chows are concealed.

Implied patterns are not counted: "Full Flush", "All Chows".

Example: 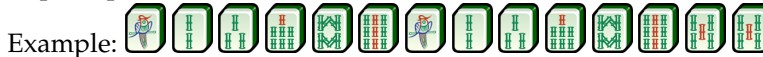

**Patterns worth 48 *fan*s**

**(14) Quadruple Chow:** A hand with four identical chows.

Implied patterns are not counted: "Pure Triple Chow", "Pure Double Chow", "Tile Hog".

Example: 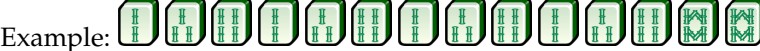

**(15) Four Pure Shifted Pungs:** A hand with four pungs or kongs in the same suit with consecutive numbers.

Implied patterns are not counted: "Pure Shifted Pungs", "All Pungs".

Example: 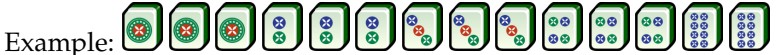

**Patterns worth 32 *fan*s**

**(16) Four Pure Shifted Chows:** A hand with four successive chows in the same suit with a fixed interval of either one or two, but not a combination of both.

Implied patterns are not counted: "Pure Shifted Chows".

Example: 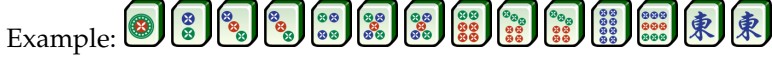

**(17) Three Kongs:** A hand with three kongs, regardless of whether they are exposed or concealed.

Implied patterns are not counted: "Melded Kong", "Concealed Kong", "Two Melded Kongs", "Two Concealed Kongs".

Example: 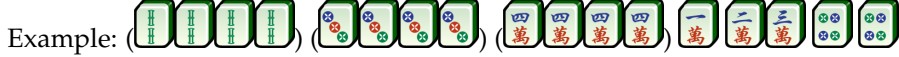

**(18) All Terminals and Honors:** A hand consisting of only terminal (number 1 or 9 of each suit) and honor tiles.

Implied patterns are not counted: "Pung of Terminals or Honors", "All Pungs".

Example: 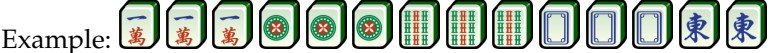

**Patterns worth 24 *fan*s**

**(19) Seven Pairs:** A hand with seven pairs. Four identical tiles can be treated as two pairs, in which case "Tile Hog" is counted.

Implied patterns are not counted: "Fully Concealed Hand", "Concealed Hand", "Single Wait".

Example: 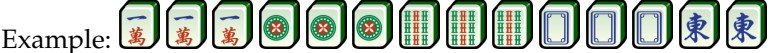

**(20) Greater Honors and Knitted Tiles:** A hand consisting of one of each honor tile, and seven tiles out of a knitted straight. A knitted straight is made up of number 1, 4, 7 of one suit, number 2, 5, 8 of second suit, and number 3, 6, 9 of a third suit.

Implied patterns are not counted: "Lesser Honors and Knitted Tiles", "All Types", "Fully Concealed Hand", "Concealed Hand".

Example: 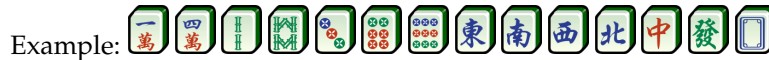

**(21) All Even Pungs:** A hand consisting of four pungs or kongs and a pair, where all tiles are suited tiles of even numbers.

Implied patterns are not counted: "All Pungs", "All Simples".

Example: ![tiles]

**(22) Full Flush:** A hand consisting of only suited tiles of the same suit.

Implied patterns are not counted: "No Honors".

Example: ![tiles]

**(23) Pure Triple Chow:** A hand with three identical chows.

Implied patterns are not counted: "Pure Double Chow".

Example: ![tiles]

**(24) Pure Shifted Pungs:** A hand with three pungs or kongs in the same suit with consecutive numbers.

Example: ![tiles]

**(25) Upper Tiles:** A hand consisting of only suited tiles of number 7, 8, and 9.

Implied patterns are not counted: "Upper Four", "No Honors".

Example: ![tiles]

**(26) Middle Tiles:** A hand consisting of only suited tiles of number 4, 5, and 6.

Implied patterns are not counted: "All Simples", "No Honors".

Example: ![tiles]

**(27) Lower Tiles:** A hand consisting of only suited tiles of number 1, 2, and 3.

Implied patterns are not counted: "Lower Four", "No Honors".

Example: ![tiles]

**Patterns worth 16 *fan*s**

**(28) Pure Straight:** A hand with three chows in the same suit of numbers 1, 2, 3; 4, 5, 6; 7, 8, 9.

Example: ![tiles]

**(29) Three-Suited Terminal Chows:** A hand with two chows of number 1, 2, 3 and 7, 8, 9 in one suit, another two chows of the same number in a second suit, and a pair of number 5 in a third suit.

Implied patterns are not counted: "All Chows", "Two Terminal Chows", "Mixed Double Chow", "No Honors".

Example: ![tiles]

**(30) Pure Shifted Chows:** A hand with three successive chows in the same suit with a fixed interval of either one or two, but not a combination of both.

Example: ![tiles]

**(31) All Fives:**   A hand with suited tiles of number five in all four melds and the pair. Implied patterns are not counted: "All Simples".

Example:

**(32) Triple Pung:**   A hand with three pungs or kongs in different suits with the same number.

Example:

**(33) Three Concealed Pungs:**   A hand with three concealed pungs or kongs.

**Patterns worth 12 *fan*s**

**(34) Lesser Honors and Knitted Tiles:**   A hand consisting of 14 tiles from these 16 tiles: number 1, 4, 7 of one suit; number 2, 5, 8 of a second suit; number 3, 6, 9 of a third suit; and all honor tiles.

Implied patterns are not counted: "All Types", "Fully Concealed Hand", "Concealed Hand".

Example:

**(35) Knitted Straight:**   A hand consisting of number 1, 4, 7 of one suit, number 2, 5, 8 of a second suit, and number 3, 6, 9 of a third suit, plus a meld and a pair.

Example:

**(36) Upper Four:**   A hand consisting of only suited tiles of number 6, 7, 8, and 9. Implied patterns are not counted: "No Honors".

Example:

**(37) Lower Four:**   A hand consisting of only suited tiles of number 1, 2, 3, and 4. Implied patterns are not counted: "No Honors".

Example:

**(38) Big Three Winds:**   A hand with three pungs or kongs of wind tiles.

Example:

**Patterns worth 8 *fan*s**

**(39) Mixed Straight:**   A hand with three chows of numbers 1, 2, 3; 4, 5, 6; 7, 8, 9 in different suits.

Example:

**(40) Reversible Tiles:**   A hand consisting of only tiles whose shape is point-symmetric regardless of colors. Available tiles are

.

Implied patterns are not counted: "One Voided Suit".

Example:

**(41) Mixed Triple Chow:**    A hand with three chows in different suits with the same numbers.

Implied patterns are not counted: "Mixed Double Chow".

Example:

**(42) Mixed Shifted Pungs:**    A hand with three pungs or kongs in different suits with consecutive numbers.

Example:

**(43) Chicken Hand:**    A hand which would otherwise be worth of zero *fan*s, with flower tiles not included.

Example:

**(44) Last Tile Draw:**    The winning hand is completed by drawing the last tile of the wall.

Implied patterns are not counted: "Self Drawn".

**(45) Last Tile Claim:**    The winning hand is completed by seizing a tile from another player when the tile wall is empty.

**(46) Out With Replacement Tile:**    The winning hand is completed when drawing a tile right after making a concealed kong or a promoted kong.

Implied patterns are not counted: "Self Drawn".

**(47) Robbing the Kong:**    The winning hand is completed by robbing the tile which another player adds to an exposed pung to form a kong.

Implied patterns are not counted: "Last Tile".

**Patterns worth 6 *fan*s**

**(48) All Pungs:**    A hand with four pungs or kongs, regardless of whether they are exposed or concealed.

Example:

**(49) Half Flush:**    A hand consisting of only honor tiles and suited tiles of one suit.

Example:

**(50) Mixed Shifted Chows:**    A hand with three consecutive chows in different suits, each shifted up one number from the one before it.

Example:

**(51) All Types:**    A hand containing tiles of all three suits, wind tiles and dragon tiles.

Example:

**(52) Melded Hand:**    Every meld and pair in the hand must be completed by tiles from other players, which means that all melds must be exposed, and one tile of the pair is the 14th winning tile from other players.

Implied patterns are not counted: "Single Wait".

**(53) Two Concealed Kongs:**   A hand with two concealed kongs.
Implied patterns are not counted: "Concealed Kong".

**(54) Two Dragon Pungs:**   A hand with two pungs or kongs of dragon tiles.
Implied patterns are not counted: "Dragon Pung".

**Patterns worth 4 *fan*s**

**(55) Outside Hand:**   A hand with terminals (number 1 or 9 of suited tiles) and honor tiles in every meld and the pair.

**(56) Fully Concealed Hand:**   A hand with no tiles from other players, which means that all melds must be concealed, and the winning tile is drawn by the winner.

**(57) Two Melded Kongs:**   A hand with two exposed kongs.

**(58) Last Tile:**   The winning tile is the last tile of the same tile that has not been revealed to all players, which means three of the same tile has been exposed in melds or discarded on the table.

**Patterns worth 2 *fan*s**

**(59) Dragon Pung:**   A hand with a pung or kong of dragon tiles. This pattern can be counted more than once if there are multiple melds meeting this requirement.
Implied patterns are not counted: "Pung of Terminals or Honors".

**(60) Prevalent Wind:**   A hand with a pung or kong of the prevalent wind of this game.
Implied patterns are not counted: "Pung of Terminals or Honors".

**(61) Seat Wind:**   A hand with a pung or kong of the seat wind of the current player.
Implied patterns are not counted: "Pung of Terminals or Honors".

**(62) Concealed Hand:**   A hand with no tiles from other players except the winning tile, which means that all melds must be concealed, while the winning tile is seized from other players.

**(63) All Chows:**   A hand with four chows and a pair of suited tiles.

**(64) Tile Hog:**   A hand with four identical tiles without forming a kong. This pattern can be counted more than once if there are multiple tiles meeting this requirement.

**(65) Mixed Double Pung:**   A hand with two pungs or kongs in different suits with the same number. This pattern can be counted more than once if there are multiple melds meeting this requirement.

**(66) Two Concealed Pungs:**   A hand with two concealed pungs or kongs.

**(67) Concealed Kong:**   A hand with a concealed kong.

**(68) All Simples:**   A hand with no terminal tiles (number 1 or 9 of suited tiles) and honor tiles.

**Patterns worth 1 *fan***

**(69) Pure Double Chow:**   A hand with two identical chows. This pattern can be counted more than once if there are multiple melds meeting this requirement.

**(70) Mixed Double Chow:**   A hand with two chows in different suits with the same number. This pattern can be counted more than once if there are multiple melds meeting this requirement.

**(71) Short Straight:**   A hand with two chows in the same suit with successive numbers, like 234 and 567. This pattern can be counted more than once if there are multiple melds meeting this requirement.

**(72) Two Terminal Chows:**   A hand with two chows in the same suit with numbers 1, 2, 3 and 7, 8, 9. This pattern can be counted more than once if there are multiple melds meeting this requirement.

**(73) Pung of Terminals or Honors:**   A hand with a pung or kong of terminal tiles (number 1 or 9 of suited tiles) or honor tiles. This pattern can be counted more than once if there are multiple melds meeting this requirement.

**(74) Melded Kong:**   A hand with one exposed kong.

**(75) One Voided Suit:**   A hand with exactly two of the three suits of tiles.

**(76) No Honors:**   A hand with no honor tiles.

**(77) Edge Wait:**   The winning tile must be a suited tile of number 3 to complete a chow of 1, 2, 3, or number 7 to complete a chow of 7, 8, 9. This pattern is not counted if there exists multiple tiles to complete a winning pattern.

**(78) Closed Wait:**   The winning tile must be a suited tile to complete a chow as the middle number. This pattern is not counted if there exists multiple tiles to complete a winning pattern.

**(79) Single Wait:**   The winning tile must be the tile to complete a pair. This pattern is not counted if there exists multiple tiles to complete a winning pattern.

**(80) Self Drawn:**   The winning tile must be drawn by the winner.

**(81) Flower Tile:**   Each flower tile is worth of one *fan*. However, the *fan* value of flower tiles cannot be counted to meet the winning criteria of 8 *fan*s. It is only given as a bonus if the player actually wins by no less than eight *fan*s without flower tiles.

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
