# Peer review of "Official International Mahjong: A New Playground for AI Research"

_algorithms, doi:10.3390/a16050235_

Round 1

Reviewer 1 Report

There is no clear research gap addressed in the paper as the paper seems to be a mere report to promote Official International Mahjong and the competition held for AI to compete against humans.

What could have been a research gap is that AI systems seem to have performed lower than humans and the need to improve the algorithms to perform better. 

There is no research contribution since the research gap was not established. This research (and its output) isn't worth considering in a journal publication.

Author Response

Thank you for your comments! As you said, our previous edition of this paper did not highlight the research gap we want to express, that AI systems cannot beat humans so far in Official International Mahjong and further research is needed to improve the algorithms. We have revised the Abstract, the sections of Introduction and Conclusion to make our logic clearer. Here we briefly describe our motivation of writing, which is conveyed in the revised version.

First of all, it is we that held the two AI competitions and the human-versus-AI competition in the paper, which is not clearly stated in the previous version and has been stressed in this version. Our original intention was to solve this game (i.e., get a near-optimal strategy to beat top humans) with advanced game AI technologies from the AI community, which we didn’t thought was that difficult. However, the results of the competitions show that modern AI algorithms including Deep Reinforcement Learning are still unable to build an AI agent to beat top humans. We realize that this game remains a challenge to AI research given its complexity, so that we try to promote it as a new benchmark. Under this motivation, this paper introduces the rules of Official International Mahjong and the competitions we have held on it. It summarizes the algorithms that have been used by the competitors, as an attempt to consolidate the results achieved on this benchmark and engage the AI community for further research.

Reviewer 2 Report

The paper is suitable for the Journal. The study presents a detailed description of the rules of Official International Mahjong, the variant used in most of official competitions due to its complex scoring rules and the emphasis on strategies. Authors illustrate the huge potential of the algorithms based on deep learning helding the AI competitions on this game, even if no AI agents are beat professional human players so far.

The paper can be accepted for publication, but a few corrections are preliminary necessary. The authors are invited to consider the following suggestions:

- The abstract should be ameliorated so that the issue, the results and main contribution for future research result clearer and more evident. It is not very scientific to express the authors' desire (We hope .....) in the Abstract. Try to avoid expressions like this also in the Introduction section (We hope this game can promote the development of AI algorithms ... .....). These sentences should be modified.

-Have authors made any improvements in the artificial intelligence applied to the game Mahjong? If yes, point this out in the article.

-The manuscript presents too many similar parts, repeating the same concepts with identic words. See for instance in the Introduction section and Section 2 Background ("Mahjong is one of the most popular games played in China, and has spread throughout the world since the early 20th century. Originated from China and with a long history dating back to the 12th century, many regional variants of Mahjong have been developed and are widely played in almost every region of China. It is estimated to have 600 million players all over the world...... These parts must figure only one time in the paper so they must be removed or totally changed when appearing in other sections.

-Adding a Flowchart at the end of the introductory paragraph, so as to give information on how the article is structured.

-Fix the citation of references in the article, as they are not arranged neatly and some are repeated.

-Some sentences don't seem to be formulated very clearly and should be changed (i. e. However, since there are too many regional variants in rules, most of the Mahjong communities focus on different regional variants and there is no such competition to cover a majority of players to have enough impart.) Authors should avoid any repetitions of words like " These algorithms include......including"

-The Conclusion section can be improved in resuming the main steps of the study carried out by pointing out the results obtained in relation to the problem examined. In addition, authors should highlight their contribution to research.

-The English language of this paper can be improved. The manuscript presents some basic grammar mistakes and imprecisions; repetitions of vocabulary and concepts. Some sentences should be written differently. You can find hereafter some sentences that need correction:

Introduction

- The strategy of Mahjong involves choosing between multiple target patterns based on the similarity between the current hand and ....

- It has fewer scoring patterns and more easier than MCR

Section 2

- The diversity of games provides a rich context for gradual skill progression to test a wide ......

- AI systems beating professional players in games with increasing complexity has always been considered ... Nowadays popular games played by humans are often much more complex, involving the factor of ....

- One important reason why these games ..... is that these games .....

- First, as a four-player game, multi-agent AI algorithms are required to solve it because both competition and cooperation exists between different agents

Section 3

- Like suited tiles, each tile have four identical copies, totaling 28 honor tiles.

- There is a special case when another player adds a fourth tile to promote a pung to a kong, a player .....

Section 4

- As an effort to promote Official International Mahjong as a playground for AI algorithms, two AI competitions were held in the year of 2020 and 2022

- Dozens of teams from universities and companies participated in the competitions, who contributed their wisdom and .......

- A human versus-AI competition was also held in the beginning of 2021 by the same organizer

- It was initially applied in the game of Bridge ......

- In each group, each four players are grouped to play 24 games of duplicate format

- Two Mahjong AI competitions have been held in the year of 2020 and 2022

- Second, each Swiss round consists of only one duplicate game with 24 games of the same till wall, but more Swiss rounds are used to ensure each agent still plays hundreds of duplicate games

- The organizers of the competitions provide a open-source library to calculate the fan value of each hand

- The results of the competition is shown in Table 4.

Section 5

- However, ....... but the fan value of the winning pattern may not enough to win

- All of these relies heavily on human expertise.

- While the design of features differ in most of the AI agents using supervised learning

- Specifically, the agents in the first Mahjong AI competition mainly uses Resnet as the CNN model

- The schemes of data preprocessing of these teams is also different from each other

- Different wind tiles can also be exchanged as long as the prevailing wind and the seat wind also changes correspondingly

Conclusions

- The most popular games played by humans have undoubtedly promoted the development of AI algorithms, in that AI systems achieving superhuman performance on these games have always been considered milestones of AI technologies

- The AI competitions held on this game indicates huge potential of the algorithms based on deep learning,

 Other minor revisions are:

- Figure 1: Distance the caption a little from the y-axis.

- Write in letters and repeat the same process for all numbers without units of measurement or where there is no formula in the text.

- Figure 4: Place it after mentioning it in the text.

- Figure 5: Place it after mentioning it in the text.

- Figure 6: Must be mentioned in the text.

Author Response

Thank you for your positive comments! We appreciate your careful and valuable suggestions on our paper and we have made our revision according to your advice. Here are our responses to your suggestions point by point.

  1. We have ameliorated the abstract to stress the competitions we have held and the results of the competitions, that algorithms based on deep learning perform well but cannot beat top human players so far, indicating this game remains a challenge to AI research. Our main contributions are also listed in bullet points in the introduction. The sentences to express our desire are modified in both abstract and introduction.
  2. No, we have not made improvement in the AI algorithms applied to Mahjong. However, we held these AI competitions and human-versus-AI competition to engage the AI community to apply modern AI algorithms on this game. Our contributions are the novel competition format (a combination of Swiss round and duplicate format) and the open-source libraries and datasets provided in our competitions to promote this game as a benchmark.
  3. We have removed or reduced these similar parts in the Introduction section and only put them in later sections such as Background.
  4. We have added a paragraph to describe the structure of this article at the end of the Introduction.
  5. Sometimes the same citation can repeat multiple times in an article when they are referred to in different sections so that readers can immediately jump to the reference when reading, but we agree with you that some citations are not arranged neatly. We have optimized the citations so that the same one can only be repeated in sections that are far apart with each other.
  6. The sentences which are not formulated very clearly have been modified. We have reconstructed the Conclusion section to make our contributions and results clearer. The contributions are also listed in bullet points in the Introduction section.
  7. We have used an English grammar software to check for grammar errors in the full paper and fixed them.

We also fixed the minor revisions you pointed out. We place these figures after mentioning them in the text, with an exception of Figure 2 which is automatically placed by LaTeX to another page because it would otherwise be placed together with Table 1 to occupy too much space of the same page. We optimize the numbers to be spelled out or not based on the following criterion.

(1) Small numbers ranging from one to ten are spelled out, while large numbers above ten are not.

(2) Numbers are always spelled out when they begin a sentence, no matter how large or small they are.

(3) When a single sentence combines small and large numbers, or numbers are frequently used in the same paragraph, all the numbers are either spelled out or written as numerals for consistency. e.g. “A full set of 144 Mahjong tiles consists of 108 suited tiles, 28 honor tiles and 8 flower tiles.”

Reviewer 3 Report

Summary: The purpose of this essay is to provide an overview of Official International Mahjong, the standard variation of the game's rules, and the AI tournaments that have been held using it.

Comments and Suggestions: 

- The authors are invited to emphasize their contributions and summarize them in bullet points in the introduction.   - The authors also need to add a paragraph describing the paper's structure at the end of the introduction.   - A figure that graphically depicts the entire proposed approach may also be included.   - Section 2 may be summarized in tabular form.   - The authors are invited to add a short paragraph about the use of formal techniques for checking the correctness of AI-based solutions.    - For this purpose, the following references may be included: 1. https://ieeexplore.ieee.org/document/9842406 2  https://dl.acm.org/doi/abs/10.1145/3503914

- It would be beneficial to share a video that explains the game's rules.

  - Is an electronic version of the game available? 
  - Figure 1 is not clear enough and needs to be enriched with more details.   - The authors need to add some preliminaries about the different AI techniques mentioned in this paper.   - It will be useful to share existing codes and datasets related to the considered problem.   - The authors need to identify the limitations of their contribution and propose more future work directions.  

Author Response

Thank you for your comments! We appreciate your valuable suggestions on our paper and we have made our revision according to your advice. Here are our responses to your comments point by point.

  1. We have restructured the Abstract, Introduction, and Conclusion to emphasize our contributions. We summarize them in bullet points in the Introduction.
  2. We have added a paragraph to describe the paper’s structure at the end of the Introduction.
  3. Our contributions are that we hold AI competitions on Official International Mahjong based on our novel competition format, summarize the algorithms the competitors used, and promote Mahjong as a new benchmark for AI research. There is not an entire proposed approach, and we have presented Figure 3 to explain our novel competition format and Table 3 and 4 to introduce the competitions.
  4. We did not use any formal techniques for checking the correctness of AI-based solutions. The judge program used in our AI competition merely implements the rules of Official International Mahjong, which terminates the game immediately when any agent takes an illegal action. No formal mathematical verification in the field of Verified Artificial Intelligence is involved in this process, so we decide not to include this topic in the article.
  5. We do not have some videos to explain the game’s rules, but a handbook is available at http://www.mindmahjong.com/adobe/MCR2021.pdf which explains the rule in detail. An electronic version is available on https://mahjongsoft.com which is an online platform for human players.
  6. The concept of information sets is introduced in Section 2.3, right before Figure 1 is referenced in the main text. In short, the number of information sets and the average size of information sets can be used to measure the state complexity of imperfect-information games. More details can be found in the citation.
  7. We have added a subsection in Section 2. to introduce some preliminaries about the different AI techniques mentioned in this paper.
  8. The existing codes and datasets can be found on the homepage of our competitions. This is conveyed in Section 4.3.

9. As the organizer of AI competitions on Official International Mahjong, the motivation of our article is to promote Mahjong as a benchmark and consolidate the results achieved on it by summarizing the current SOTA algorithms. Since we are not proposing a new approach to solve an existing problem, it is difficult to say what the limitations could be. The results of human-versus-AI competition shows that current Mahjong AI agents still cannot beat professional human players, which is where the research gap lies. We want to engage the AI community for further research, so we do not include possible future work directions in this article, but only propose this benchmark for the AI community.

Round 2

Reviewer 3 Report

The authors conisdered my comments and suggestions. Good luck.